# Research on the Monitoring Method of the Refuse Intake Status of a Garbage Sweeper That Is Based on the Synergy of a Wind Speed Sensor and an Ultrasonic Sensor

**DOI:** 10.3390/s25134010

**Published:** 2025-06-27

**Authors:** Zihua Chen, Qingbing Zeng, Zhongwen Chen, Yixiao Zhang, Heng Yang

**Affiliations:** 1School of Mechanical and Automotive Engineering, Shanghai University of Engineering Science, Shanghai 201620, China; m315124339@sues.edu.cn (Z.C.); m315124308@sues.edu.cn (Z.C.); m315124237@sues.edu.cn (Y.Z.); 2Hunan Niu En Chi New Energy Vehicle Co., Ltd., Zhuzhou 412007, China; chenhui8044@163.com

**Keywords:** garbage sweeper, overflowing rubbish storage compartments, vacuum duct clogging, ultrasonic sensor, wind speed sensor

## Abstract

Garbage sweepers are crucial to municipal cleaning; however, they frequently encounter two significant challenges during their operations: overflowing rubbish storage compartments and vacuum duct clogging. The conventional method of monitoring overflowing refuse and clogging vacuum ducts is based on manual labour, which is inefficient and susceptible to error. Consequently, this investigation suggests an automated monitoring approach that is predicated on the integration of wind speed sensors and ultrasonic sensors. The ultrasonic sensors assess the blockage status by monitoring the height of the accumulation of rubbish in real time, while the wind speed sensors monitor the change in wind speed. The experimental results indicate that the system is an effective solution for monitoring the refuse intake status of intelligent sweepers, significantly reducing manual intervention and improving the operational efficiency of the equipment. Additionally, the system is accurate and reliable.

## 1. Introduction

The introduction of waste sweepers has significantly enhanced the cleaning efficiency and has become a critical instrument for modern urban environmental management, as the cleaning of urban waste has become increasingly burdensome due to the acceleration of urbanisation [1,2,3,4]. The rubbish sweeper, at present, is primarily dependent on manual inspection to assess the overflowing state of the rubbish storage compartment and the clogging situation of the vacuum pipe during the operation process. Nevertheless, it will encounter the two primary challenges of overflowing rubbish storage compartments and clogging vacuum pipes. This method is complex, which not only affects the cleaning efficiency but also exposes the staff to potential safety hazards, such as slipping, collapsing, or contracting harmful pathogens [5]. Consequently, more effective methods are required to monitor the status of the rubbish intake.

### 1.1. Investigation of Overflow Monitoring Technology

Currently, waste overflow monitoring technology is primarily utilised in fixed rubbish bins (e.g., community or public rubbish bins), and the primary monitoring methods are infrared monitoring [6], visual monitoring [7,8], and ultrasonic monitoring [9]. While infrared and visual monitoring methods can satisfy the requirements of fixed bin overflow monitoring, they are problematic for rubbish sweeper overflow monitoring due to the vibration of the sweeping process, as well as the low light, sand, dust, and leaves in the rubbish storage bin. The visual monitoring technique is susceptible to contamination or obstruction by the camera in this environment, and the image will be blurred as a result of the rubbish sweeper’s vibration as well. The infrared monitoring technique is limited in its ability to ascertain whether the rubbish is overflowing or not, and it is unable to continually monitor the height of the rubbish pile.

Ultrasonic sensors offer clear benefits in the context of rubbish sweepers. They are capable of withstanding dust, water mist, and other adverse environmental interference by transmitting high-frequency acoustic waves and receiving reflective signals from the media interface, thereby achieving consistent monitoring of the height of rubbish accumulation [10]. Furthermore, the visual monitoring method necessitates a high frame rate camera and embedded image processor for cost-effectiveness, whereas the ultrasonic monitoring method requires only ultrasonic sensors to monitor rubbish overflow, which is a clear advantage in terms of cost.

### 1.2. Technology Research for the Monitoring of Dust Extraction Duct Blockage

The transient pressure wave reflection method, which is based on the propagation characteristics of pressure waves, is the primary method used in existing pipeline clogging monitoring techniques. This method captures the reflected signals of pressure waves within the pipeline system to achieve clogging localisation and condition assessment [11]. The second method is the fibre optic sensing technique, which uses distributed fibre optic sensors to monitor the changes in the pipeline’s temperature [12].

The wind speed sensor has significant cost advantages in the monitoring of hoover duct clogging in refuse sweepers when contrasted with the transient pressure wave reflection method and fibre optic sensing technology. The transient pressure wave reflection method necessitates the use of high-frequency pressure sensor arrays and specialised signal processors, while the fibre optic sensing method is relatively expensive. In contrast, the wind speed sensor has a straightforward structure, a straightforward installation process, and fewer maintenance requirements, which significantly reduces the operating costs of monitoring the clogged vacuum ducts of rubbish sweepers.

## 2. Architecture of the System

To facilitate the simultaneous real-time surveillance of vacuum duct blockage and waste fullness, this waste sweeper employs a combination of wind speed sensors and ultrasonic sensors. The ultrasonic sensor is installed on the upper cover of the rubbish storage compartment. This decision was made after a thorough analysis of the airflow within the compartment and the permissible installation locations. The impeller-type wind speed sensor is employed for real-time monitoring of wind speed changes, and the wind speed sensor is embedded in the air outlet of the ventilation pipe, based on the analysis of the most sensitive location of the wind speed when the sweeper is blocked and the possible locations allowed for installation in the rubbish storage compartment. This installation position mitigates the risk of sensor contamination and physical damage, extends the equipment’s service life, and prevents direct contact with the refuse. Figure 1 illustrates the ultrasonic sensor and wind speed sensor’s implementation locations.

## 3. Detailed Technical Information

The technical specifications of the ultrasonic sensor and the wind speed sensor will be described in this chapter after the system architecture has been described.

### 3.1. Ultrasonic Sensors’ Technical Specifications

#### 3.1.1. Technical Parameters and Design of Ultrasonic Transducers

The ultrasonic sensor operates on a modified Time of Flight (ToF) measurement method. The sensor hardware configuration employs a layered architecture: raw data is gathered by ultrasonic sensors, subsequently pre-processed by a signal conditioner (comprising signal amplification and filtering circuits), and ultimately transmitted to the data processing platform for the execution of multi-level filtering algorithms. This architecture mitigates environmental interference and enhances data reliability. The sensor is composed of a transmitter and a receiver that produce ultrasonic vibrations with a frequency of f. These pulses are transmitted through the atmosphere and are reflected to the receiver upon contact with the surface. The system determines the distance d between the sensor and the debris surface by measuring the time interval t between the transmission and reception of the sound waves, in conjunction with the propagation speed of the sound waves in the air, c. The distance is determined by employing the following formula:(1)$$d=\frac{c\cdot t}{2}$$

The round-trip duration of the sound wave is denoted by *t*, while the propagation speed of the sound wave in air is represented by *c* in Equation (1). Because the sound wave must traverse a round-trip path for emission and return, the actual distance is only half of the total distance travelled [13]. This approach has a broad spectrum of applications in the field of non-contact ranging, including obstacle monitoring and robot navigation [14].

The ambient temperature *T* has a substantial impact on the acoustic propagation velocity *c*. The relationship can be articulated as follows:(2)$$c=c_{0}+K\cdot T$$

In Equation (2), where *c*_0_ is the speed of sound at the reference temperature (typically 0 °C), *K* is the temperature coefficient and *T* is the ambient temperature. A high-precision temperature sensor is integrated into the system to guarantee measurement accuracy. This sensor dynamically modifies the value of *c* and monitors *T* in real time. The system effectively eliminates the influence of temperature fluctuation on ranging accuracy and enhances the robustness of monitoring through this temperature compensation mechanism. The reliability of ultrasonic measurements has been improved through the implementation of a comparable methodology in transformer partial discharge monitoring [15].

The operational frequency range of standard ultrasonic sensors typically spans from 40 to 60 kHz, with a measuring range of 0.02 m to 5.0 m [13]. The output interface accommodates several modes, including automatic/controlled UART, PWM, switch quantity, and RS485, which supports the MODBUS-RTU protocol. The data acquisition rate, defined as the frequency at which the system collects sensor readings per unit time, can attain 2 Hz to facilitate real-time monitoring.

#### 3.1.2. Data Processing for Ultrasonic Sensors

The echo signal amplitude *V* of the pulse emitted by the ultrasonic transducer during propagation can be represented as follows:(3)$$V=\frac{P_{0}\cdot e^{-\alpha d}}{d^{2}}\cdot cos\theta$$

Equation (3), where *P*_0_ represents the initial transmit power, *α* represents the air attenuation coefficient, *d* represents the monitoring distance, and *θ* represents the tilt angle of the debris surface. The formula demonstrates that the echo strength *V* is decreasing exponentially as the distance *d* increases. The signal strength is considerably reduced when the tilt angle *θ* of the rubbish surface is greater, which may result in monitoring errors. In the meantime, the ultrasonic signal exhibits non-smooth characteristics in the monitoring of rubbish overflows. Firstly, the rubbish surface generates irregular scattering as a result of loose accumulation, resulting in a high-frequency random fluctuation of the echo signal. Secondly, the mechanical vibration of the sweeper vehicle introduces impulsive outliers, and the difference in the composition of the rubbish triggers time-varying attenuation [16]. To improve the precision of fullness monitoring, the outlier filtration algorithm must possess multi-stage processing capabilities to accommodate these composite noise features.

The field of signal optimisation has extensively synergistically employed multi-stage filtering algorithms. In the study of rigid body inertia parameter estimation, Kim et al. (2016) [17] innovatively combined extended Kalman filtering (EKF) with Savitzky–Golay (SG) filtering to achieve dynamic noise suppression through EKF and SG filtering to complete high-frequency noise smoothing. This resulted in an inertia matrix estimation error of less than 1%. In 2019, the Sivagami team constructed a meteorological prediction model, and the prediction mean-square error was reduced by Kalman filtering after SG filtering efficiently smoothed the temperature time-series fluctuations. This further confirmed the synergistic advantage of SG filtering and Kalman filtering [18]. Zeng et al. (2024) [19] introduced a joint IQR-SG algorithm in the industrial drilling sector. This algorithm effectively reduced the acceleration calculation error by eliminating abnormal vibration signals using the quadratic distance method, in conjunction with SG filtering to preserve the trend characteristics of the formation penetration rate. The multipath effect interference was effectively eliminated in the optimisation of the interior positioning system by the cascade of IQR and Kalman filtering, as discovered by Mahardhika et al. (2021) [20]. The application can effectively eradicate multipath effect interference and enhance the positioning accuracy to the MSE level of 1.09 m^2^ [20]. These research findings serve as a methodological guide for the hierarchical processing of composite noise.

The ultrasonic sensor’s minimum measuring distance is d_min_, which is the value at which the reading is fixed when the distance of debris or interfering objects from the sensor is less than or equal to d_min_. In practice, the sensor may emit the minimum measurement distance value d_min_ in response to temporary obstructions, such as leaves that are stuck or floating objects that interfere. Consequently, if the ultrasonic sensor continues to output the minimum detection distance value d_min_ after the multi-stage anomaly filtering algorithm has preprocessed the sensor data, there are two potential scenarios: (1) the height of the rubbish accumulation has exceeded the physical tolerance value (overflow state) or (2) transient interference caused by floating rubbish in the compartment or foreign objects on the surface (pseudo overflow signal). The accuracy of overflow monitoring can be enhanced by incorporating an overflow recognition algorithm on top of multi-stage filtration to accurately differentiate between the aforementioned cases.

The hybrid filtering idea of multilevel filtering and state judgement has been applied in the field of signal processing, for example, Li et al. proposed a noise suppression method based on a hybrid noise estimation technique [21]. This method effectively improves the signal processing accuracy under the complex noise environment (e.g., inside the car) by combining the multi-channel and single-channel estimation techniques. The method verifies its accurate estimation ability for local noise in the experiments. These studies provide a good reference for the proposal of sensor filtering algorithms for rubbish sweepers.

The sensor data is processed through a three-stage progression of anomaly dynamic thresholding, nonlinear trend maintenance smoothing, and state space optimal estimation, and is formed into accurate data after passing through the overflow identification algorithm. The multistage anomaly filtering algorithm proposed in this study consists of Dynamic IQR-based Anomaly Detection, Savitzky–Golay Time-Frequency Denoiser, and Enhanced Kalman State Tracker. Figure 2 illustrates the algorithm for processing sensor data that was suggested in this investigation.

Initially, the Dynamic IQR-based Anomaly Detection method accomplishes robust monitoring of anomalies by calculating the statistical distribution range of the data. The effective range is defined as:(4)$$d_{r}=[Q_{1}-1.5\cdot IQR,Q_{3}+1.5\cdot IQR]$$

Equation (4): *IQR* = *Q*_3_ − *Q*_1_. In comparison to standard deviation methods, *IQR* is more adaptable to non-Gaussian distributed data and eliminates impulse noise [22].

This is succeeded by the Savitzky–Golay Time-Frequency Denoiser, an SG filter that is based on a polynomial least-squares fit within a sliding window. The centroid filter value of this filter is determined by the following equation:(5)$$d_{i}=\sum_{j=-m}^{m}c_{j}\cdot x_{i+j}$$

In Equation (5), *c_j_* is the precomputed convolution coefficient (window 9 points, 3rd order polynomial). The SG filter has been demonstrated to reduce high-frequency noise while maintaining the signal’s peak characteristics [23].

The Enhanced Kalman State Tracker is the subsequent step, in which Kalman filtering integrates the system model with the observed data through a prediction-update mechanism.(6)$$\begin{array}{l} \text{Predict:}\;{\hat{d}}_{k}^{-}={\hat{d}}_{k-1},P_{k}^{-}=P_{k-1}+Q\\ \text{Update:}\;K_{k}=\frac{P_{k}^{-}}{P_{k}^{-}+R},{\hat{d}}_{k}={\hat{d}}_{k}^{-}+K_{k}(z_{k}-{\hat{d}}_{k}^{-}) \end{array}$$

Kalman filtering has been demonstrated to be an algorithm that is capable of estimating the state of linear dynamic systems. Kalman filtering can generate more precise estimates in dynamic environments by integrating data from sensors [24].

Lastly, the full overflow recognition algorithm, as illustrated in Figure 2, initiates the timing analysis upon the sensor reading reaching the d_min_ threshold. It then calculates the mean value by analysing the first N sets of data at the current instant.(7)$$d_{a}=\frac{1}{N}\sum_{i=1}^{N}d_{t-i}$$

If d_a_ − d_min_ ≥ d_l_ (the specific value of d_l_ is contingent upon the structure of the rubbish truck box and the installation height of the sensor), it is determined to be a transient interference (e.g., obscured by a floating object). The system enters a locking state when it is determined to be a transient interference and is continuously determined not to overflow for the time t_l_. The hysteresis effect in the control system is simulated by determining the locking time based on the actual operational scenarios of the rubbish truck (e.g., pipe blockage or driver’s rest). The literature [25] demonstrates that this state machine logic is particularly well-suited for signal processing in dynamic environments and can substantially reduce false alarms by managing state transitions. If the system is not in a locked state and d_a_ − d_min_ < d_l_, it is determined to be close to complete overflow, and an alarm is triggered. This sliding window averaging improves data stability by normalising short-term fluctuations, and the literature [26] suggests that it is effective in evaluating the impact of noise on time series analysis.

### 3.2. Technical Specifications for Wind Speed Sensors

#### 3.2.1. Technical Parameters and Wind Speed Sensor Design

The wind speed sensor utilises high-precision sensing technology in conjunction with an impeller-type design to achieve real-time wind speed monitoring in a dynamic environment when monitoring the clogging of the vacuum conduit of the rubbish sweeper.

The impeller wind speed sensor operates based on the coupling relationship between fluid dynamics and the Hall effect. The Hall effect module generates a pulse signal by monitoring the magnetic pole change in the impeller when the airflow in the duct causes the impeller to rotate. The frequency of this signal, *f*, is proportional to the impeller speed, *n*:(8)$$n=\frac{f}{N}$$

In Equation (8), the frequency is denoted by *f*, and the number of pulses generated per revolution of the impeller is denoted by *N*. This number is dictated by the magnetic pole layout. The theoretical distance of a single revolution to propel the airflow is determined by wind tunnel experiments, which calibrate the effective wind range *L* corresponding to each revolution of the impeller. When the rotational speed and effective wind range are combined, the theoretical wind speed can be expressed as follows:(9)$$v=n\cdot L=\frac{f\cdot L}{N}$$

A calibration coefficient *k* is introduced to account for the effects of fluid properties (e.g., turbulence, air density) and mechanical losses. The final wind speed equation is as follows:(10)$$\nu =k\cdot \frac{f\cdot L}{N}$$

To guarantee the precision of the sensor’s measurements in dynamic environments, the calibration coefficient *k* must be determined experimentally or from calibration curves supplied by the manufacturer.

In practical applications, standard impeller-type wind speed sensors possess a range of 0–20 m/s, a start-up wind speed threshold of roughly 0.3 m/s, and provide analogue (0–10 V) or digital (RS485) signal outputs. The data acquisition rate, defined as the number of wind speed samples collected per second, attains 1–10 Hz to accommodate various operational situations.

#### 3.2.2. Data Processing for Wind Speed Sensors

In this investigation, the clogging status of the dust-absorbing duct is monitored in real-time by measuring the change in wind speed at the outlet of the ventilation duct using a wind speed sensor. To achieve automated hierarchical monitoring of the degree of clogging, a timing analysis algorithm is developed that combines the initialisation stage optimisation, sliding-window averaging, and a multi-stage thresholding strategy to enhance the accuracy of the clogging monitoring of the vacuuming duct. Figure 3 illustrates the algorithm’s precise implementation flow.

To prevent the transient wind speed fluctuation during turbine acceleration from triggering misjudgment and to prevent false alarms during the startup phase by setting the initialisation phase, no data collection is conducted during the t_0_ time after system startup in Figure 3, drawing on the concept of hysteresis effect in control systems [27].

For each sampling point *V*(*t*), the system calculates the mean value of wind speed within a movable window of length L_a_.(11)$$v_{a}=\frac{1}{L}\sum_{k=0}^{L-1}v(t-k)$$

By normalising short-term wind speed fluctuations, this step improves data stability while maintaining sensitivity to persistent wind speed decreases. This study effectively balances real-time and accuracy through the use of sliding windows, which can be effectively employed for noise filtering in time series analysis [26].

The system implements a three-level threshold for judging the blockage state in accordance with the sliding window average wind speed v_a_. The minimum wind speed in the smooth state of the pipeline is denoted by v_n_, while the minimum wind speed in the partially blocked state is denoted by v_p_ (as determined by experiments).

Normal state: The green indicator is activated when the system determines that v_a_ ≥ v_n_ is a normal working state.

Partial blockage (the presence of foreign matter accumulation in the pipeline results in a 30–70% reduction in airflow cross-sectional area, and the suction power is significantly reduced but not completely blocked): the system determines that it is a moderate blockage when v_p_ ≤ v_a_ < v_n_, and it initiates a low-frequency beeping (1 Hz) to prompt drivers to check the status of the pipeline. A yellow warning is triggered.

Complete blockage: The system detects a severe blockage when v_a_ < v_p_, activates a red emergency alarm (3 Hz), and automatically disconnects power to the compressor to safeguard the equipment from overload damage.

The multilevel decision-making mechanism of the composite filtering algorithm in the literature [28] is referenced by these threshold settings. This mechanism substantially reduces the false alarm rate by integrating wind speed magnitude analysis with time persistence.

The automated graded warning of the clogging status of the dust vacuum pipe is achieved through the time sequence analysis algorithm of the wind speed sensor. The system is capable of triggering a low-frequency alarm at the early stage of blockage, prompting the driver to clean up promptly, and cutting off the fan power supply immediately at the time of complete blockage, preventing the equipment from failing due to overload, and resulting in the accurate monitoring of pipe blockage, based on the sliding window average calculation and three-level threshold response mechanism (green light normal, yellow light warning, red light emergency shutdown).

## 4. Results of the Experiment

### 4.1. Experiment with Monitoring Overflow

A practical test verification was conducted at the actual sweeping site to confirm the efficacy of the refuse fullness monitoring. The experimental scene, as illustrated in Figure 4, replicates the actual operating environment. A mixed refuse layer (65% dead leaves, 25% soil, and 10% plastic debris) is applied to the test site, which effectively simulates the impact of floating rubbish on the sensor. The experiment employed the DYP-A13-V1.0 ultrasonic sensor manufactured by Shenzhen Dianyingpu Technology Co., Ltd., featuring an operational frequency of 40 kHz, a supply voltage range of DC 3.3–24 V, and adherence to the IEC61000-4-2 electrostatic protection standard. An RS485 communication interface (Baud Rate: 9600 bps; Data Bit: 8 bits; Stop Bit: 1 bit) is utilised by the ultrasonic sensor, which has a range of 0.25–2.0 m. It is installed on the upper cover of the rubbish storage compartment and has a protection level of IP68, allowing it to withstand the high humidity and dust environment of the compartment. The sensor’s raw data was within the 1000–1050 mm range in the empty compartment, and the entire test procedure, from empty to overflowing, lasted approximately 2200 s.

This study contrasts the processing effects of various data processing algorithms to systematically verify the cooptimisation effect of multimodal filtering algorithms. First, Figure 5 illustrates the single algorithm processing effect test, and the comparison test results are as follows:

Raw Data Characteristics: The ultrasonic sensors’ trash distance data exhibited substantial non-stationary fluctuations over 2200 s, with pulse-type anomalies coexisting with high-frequency noise.

The IQR filtering effect: The IQR filtering effect is a dynamic threshold cleaning method that uses quartile spacing to eliminate large outlier pulses and enhance data smoothing by 30% (from a standard deviation of 18.7 mm to 13.1 mm). However, it is unable to suppress low-frequency trend noise.

SG filtering effect: the high-frequency noise power spectral density is reduced by 15 dB (green line smoothing RMS = 4.2 mm) in the context of a windowed 9-point, 3rd-order polynomial fit. However, there is still a residual high-frequency noise.

Kalman filtering effect: the standard deviation of random noise is reduced to σ = 2.3 mm (42% better than SG filtering) through the dynamic tracking of the state-space model. However, it is still sensitive to abrupt anomalies, and the local residual amplitude is a significant factor.

Subsequently, a comparison test of multi-algorithm cooptimization algorithms was implemented, as illustrated in Figure 6. The results of the comparison test are as follows:

SG-Kalman cascade: The residual amplitude is diminished by inputting SG pre-smoothing (which eliminates high-frequency noise) into Kalman filtering. Nevertheless, the anomaly leakage rate reaches 38.2% as a consequence of the fixed-window property of SG filtering, which causes sudden anomalies to be over-smoothed.

IQR-SG cascade: Data stability (σ = 7.5 mm) is increased by SG fitting after IQR anomaly cleansing, compared to single IQR filtering (σ = 13.1 mm). However, the transient response overshoot remains higher.

IQR-Kalman cascade: Kalman dynamic tracking after IQR preprocessing, with improved impulse noise rejection (83% fewer anomalies) but increased sensitivity to trend noise (residual drift rate 0.8 mm/s).

IQR→ SG→ Kalman three-level cascade strategy (the algorithm proposed in this thesis): the three-algorithm fusion architecture achieves noise suppression, enhances monitoring accuracy, reduces high-frequency noise power by 26 dB, and eliminates pulse anomalies.

This study compares the performance of each module when running individually and in combination, and quantifies the data standard deviation and impulse noise rejection rate of each module to verify the necessity of the three-level cascade algorithm. The impulse noise rejection rate *η* is defined by the equation.(12)$$\eta =\frac{N_{0}-N}{N_{0}}\cdot 100\%$$

In Equation (12), *N*_0_ represents the number of bursty deviations in the original signal and *N* represents the number of residual anomalies following algorithmic processing. Table 1 illustrates the performance comparison of each algorithm and algorithm combination.

This experiment confirms that the three-level cascade architecture of IQR, SG, and Kalman outperforms any single-module or dual-module combination in both noise suppression and dynamic tracking through complementary synergies. Additionally, it offers an optimal solution for overflow monitoring in complex environments of rubbish trucks.

The system accurately activates the overflow alarm in numerous real overflow condition tests using the 10 s window mean analysis (d_a_ = 256 mm, d_a_ − 250 < 10 mm). Figure 7 illustrates the rubbish overflow, which further confirms the algorithm’s reliability.

### 4.2. Experiments to Monitor the Obstruction of Dust Extraction Ducts

To confirm the wind speed sensor’s reliability in monitoring duct blockage, this investigation implements a multi-case comparison experiment. The experiment utilised a stainless steel impeller wind speed sensor produced by Hebei Feimeng Electronic Technology Co., Ltd., Handan, China. The wind speed sensor is an impeller-type device that is installed at the outlet of the ventilation duct. It has a range of 0–20 m/s and a start-up wind speed of 0.3 m/s. The device is protected by an IP67 rating, operates on a DC 12–24 V power supply, and has a working current of approximately 20 mA (12 V power supply). It communicates using the RS485 protocol (Baud Rate: 9600 bps; Data Bit: 8 bits; Stop Bit: 1 bit). The wind speed sensor hardware system employs a three-tier architectural design: the sensing layer utilises an embedded industrial microcomputer processor and an optoelectronic conversion mechanism to directly digitise the impeller pulse signal; the transmission layer employs shielded cables to connect to the signal conditioner, which incorporates electrostatic protection and signal isolation circuits to ensure stable signal transmission; the processing layer receives the conditioned digital signal from the data processing platform and executes a sliding window algorithm and multi-level threshold analysis to facilitate real-time blockage grading early warning. The garbage will be manually piled to create various stacking forms during the experiment to replicate the pipe blockage scenario that occurs during the sweeper’s actual operation. To begin, simulate the multi-stage blockage condition of the dust suction pipeline in a controllable manner by allowing the rubbish to be distributed as in the experiment in Figure 4. Subsequently, allow the rubbish sweeper to operate normally after it enters steady state operation. This will be achieved by dynamically stacking the rubbish load density in the designated area of the path. Reference Figure 8 for an illustration.

Figure 8 illustrates that the wind speed data of the system during the initialisation phase (0–12 s) increases from 0 m/s to 6 m/s within a 12 s timeframe. Currently, the system will generate a false alarm if only the threshold strategy is implemented. Consequently, this system implements a 10 s initialisation phase (t_0_ < 10 s). The algorithm’s determination is effectively mitigated by the introduction of the initialisation phase, which prevents the influence of the start-up phase. At 58–71 s in Figure 8, the wind speed is consistently between 4 m/s and 5 m/s as the sweeper sweeps the small-density rubbish road surface for approximately 50 s, dynamically superimposing the rubbish load density in the preset area of the path. At this time, the pipeline is partially blocked. The yellow warning is precisely initiated during this phase by the system, which calculates the mean value of the 10 s sliding window (v_a_ = 4.5 m/s). The engineering reasonableness of the threshold setting was confirmed by the fact that the height of garbage accumulation in the pipe was 42% of the diameter of the suction pipe (measured value) at this time.

When the duct is entirely obstructed, the wind speed remains within the range of 1 m/s to 3 m/s from 71 s to 86 s. The present algorithm reduces the risk of equipment overloading through hardware linkage by calculating the mean value of the 10 s sliding window (v_a_ = 2.2 m/s) and cutting off the power supply to the fan within 10 s. This phase initiates a red alarm.

In this investigation, a timing analysis algorithm that integrates initialisation phase optimisation, sliding window averaging, and a multi-level thresholding strategy is implemented. The computational complexity of the algorithm is reduced tenfold compared to sliding window averaging. The response time for device protection is reduced as a consequence of the use of a three-level state machine decision mechanism and hardware linkage design.

The experimental results demonstrate that the system achieves high accuracy in waste overflow monitoring through fusion algorithms. In vacuum pipe clogging monitoring, the multilevel threshold algorithm, in conjunction with 10 s sliding-window mean analysis, reduces the response time for complete clogging to 10 s, thereby verifying the system’s reliability in intricate sanitation scenarios.

## 5. Conclusions and Discussion

### 5.1. Discussion

The monitoring system described in this work shows commendable performance in experiments; nonetheless, certain areas merit further investigation regarding its practical use.

The expense of system deployment is a significant factor. The total hardware cost of a single vehicle system, utilising selected industrial-grade sensors (ultrasonic sensors priced at approximately $15–40 per unit and wind speed sensors at about $55–80 per unit) and embedded computing units, can be maintained within the range of $140–275. The real-time execution of the three-stage filtering and state decision algorithm on a standard embedded processor (e.g., ARM Cortex-M4) incurs a computational overhead of approximately 1.2 ms per sampling point and an energy expenditure of roughly 5.4 mW (excluding the sensor’s inherent power consumption), which exerts a manageable influence on the current battery system and computational resources of the sweeper. The anticipated annual maintenance expenses, encompassing sensor cleaning, calibration, and possible replacement, are projected to range from 10% to 15% of the hardware cost. In relation to the total expense of the garbage sweeper, this additional cost constitutes a minor fraction, and it is anticipated to yield benefits by diminishing manual inspections, preventing equipment malfunctions, and enhancing efficiency.

The primary impediment to adoption is the considerable complexity of real-world sanitation situations. Despite the selected sensors being well-established in the industrial sector and demonstrating reliability in simulated field trials, they encounter obstacles in actual operations, including the potential for sensor contamination, interference from harsh working conditions, and variations in system adaptation. In the future, comprehensive and prolonged field performance evaluations will be necessary in real urban road operational contexts.

The analysis indicates that, within the current algorithm framework, the chosen sensor specifications (including 40 kHz ultrasonic, 2 Hz sampling rate, and impeller wind speed sensor) adequately fulfil monitoring requirements while achieving an optimal balance among cost, environmental adaptability, and performance. Pursuing greater standards, such as enhanced frequency/precision ultrasonic sensors or expedited response wind speed sensors, may yield minimal marginal gains while increasing costs. Future optimisation should prioritise adaptive enhancements at the algorithmic level.

### 5.2. Conclusions

This study integrates the benefits of ultrasonic sensors and wind speed sensors with innovative algorithms to effectively mitigate the effects of sand and dust interference in the closed chamber, leaf floating, vibration noise, and poor lighting conditions on the monitoring effect. The goal is to ensure that the monitoring system for overflow monitoring of the rubbish sweeper and the clogged vacuum pipe has good monitoring accuracy, robustness, and equipment protection performance. The study adds, and field experiments demonstrate, that the proposed method is highly accurate in monitoring the overflow of refuse and provides precise warnings of pipeline blockages. The primary focus of the present investigation is the validation of the road simulation environment. Subsequently, comprehensive and long-term field performance tests should be conducted in actual operating scenarios on urban roads.

## Figures and Tables

**Figure 1 sensors-25-04010-f001:**
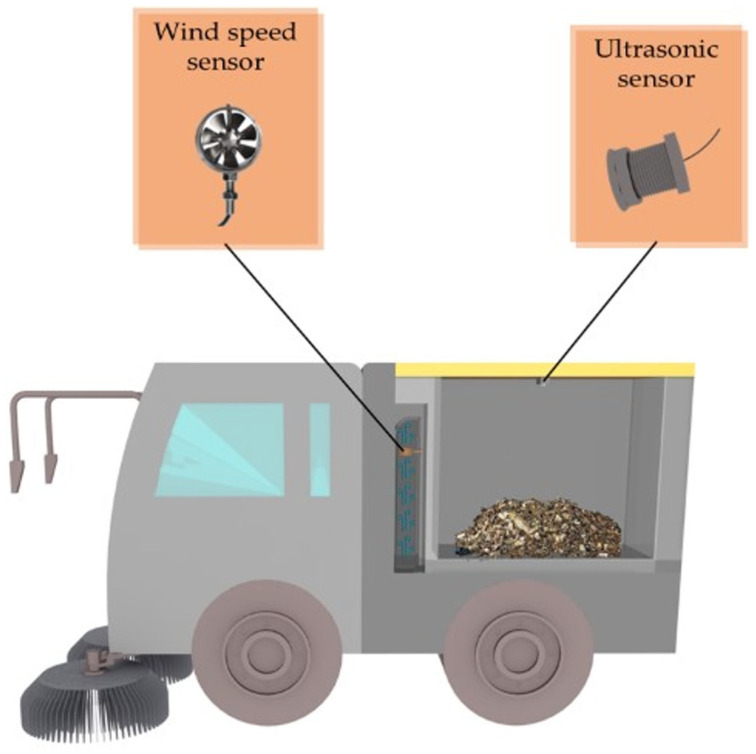
System installation schematic.

**Figure 2 sensors-25-04010-f002:**
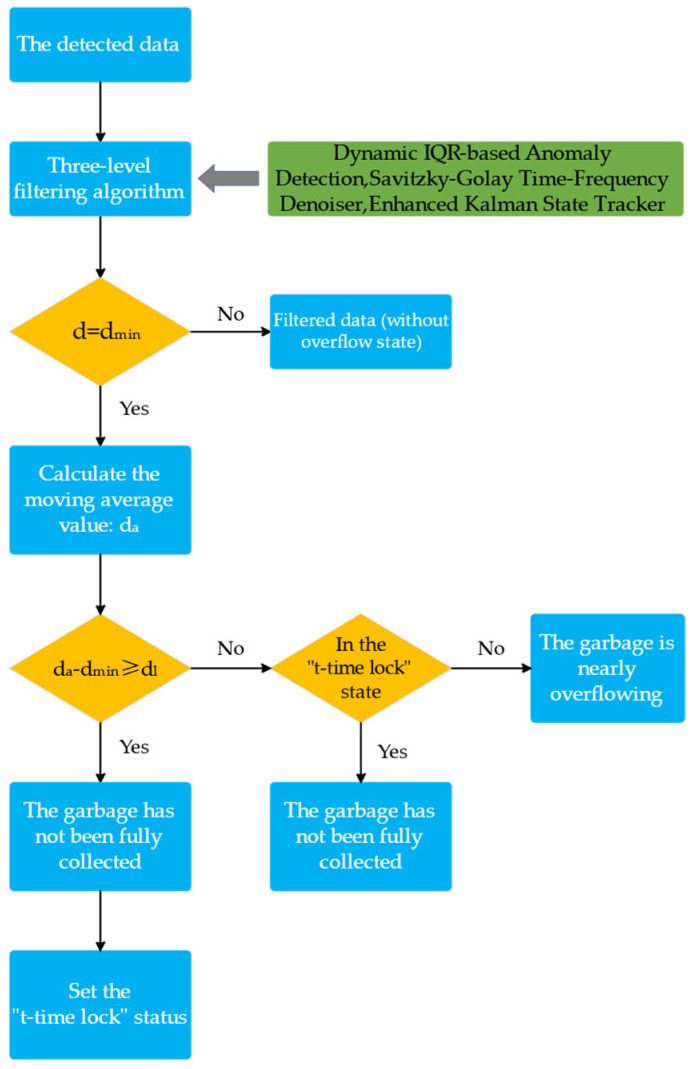
Ultrasonic data processing algorithm flowchart.

**Figure 3 sensors-25-04010-f003:**
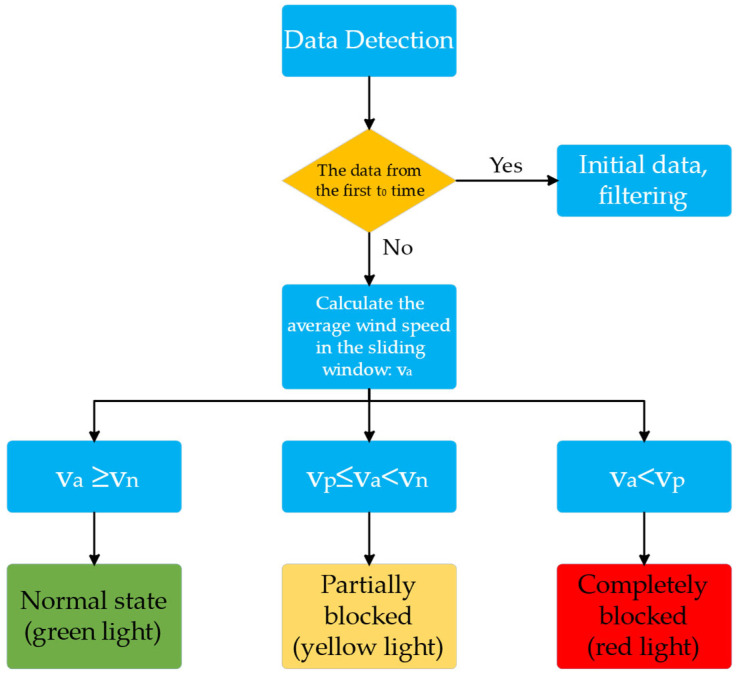
Algorithm for processing wind speed data flow chart.

**Figure 4 sensors-25-04010-f004:**
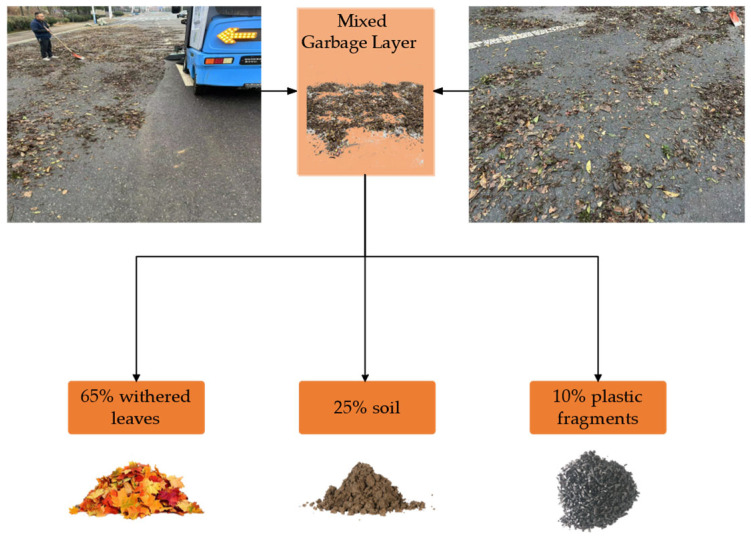
Test site for overflow monitoring.

**Figure 5 sensors-25-04010-f005:**
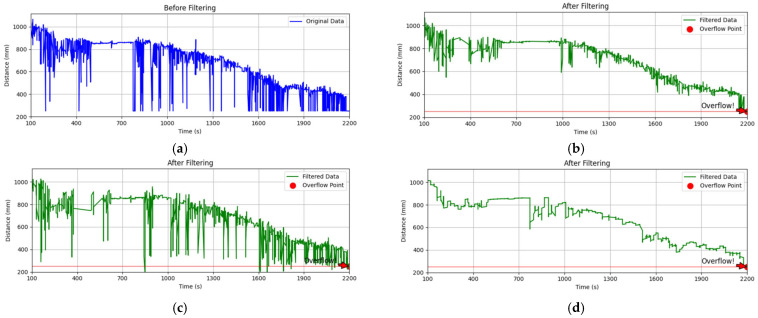
(**a**) Original data plot; (**b**) IQR filtering effect; (**c**) SG filtering effect; (**d**) Kalman filtering effect.

**Figure 6 sensors-25-04010-f006:**
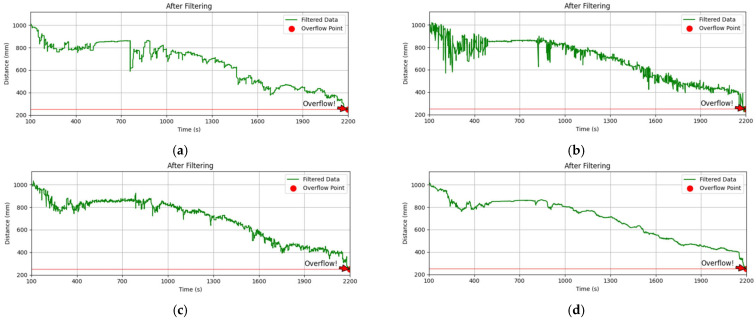
(**a**) The filtering effect of the algorithm in this thesis. (**b**) The IQR-SG cascade filtering effect. (**c**) The IQR-Kalman hybrid. (**d**) The SG-Kalman cascade filtering effect.

**Figure 7 sensors-25-04010-f007:**
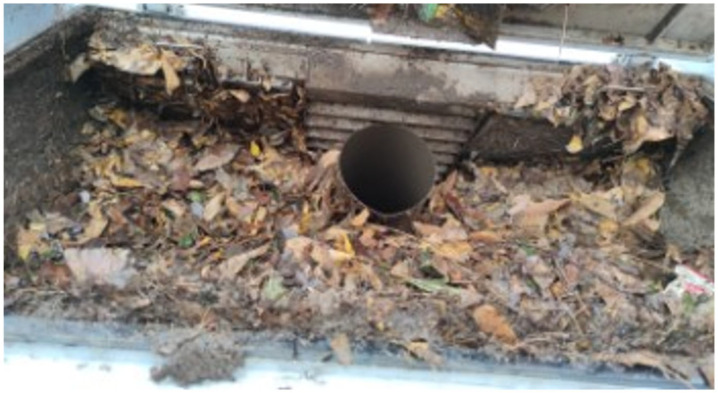
Refuse accumulation.

**Figure 8 sensors-25-04010-f008:**
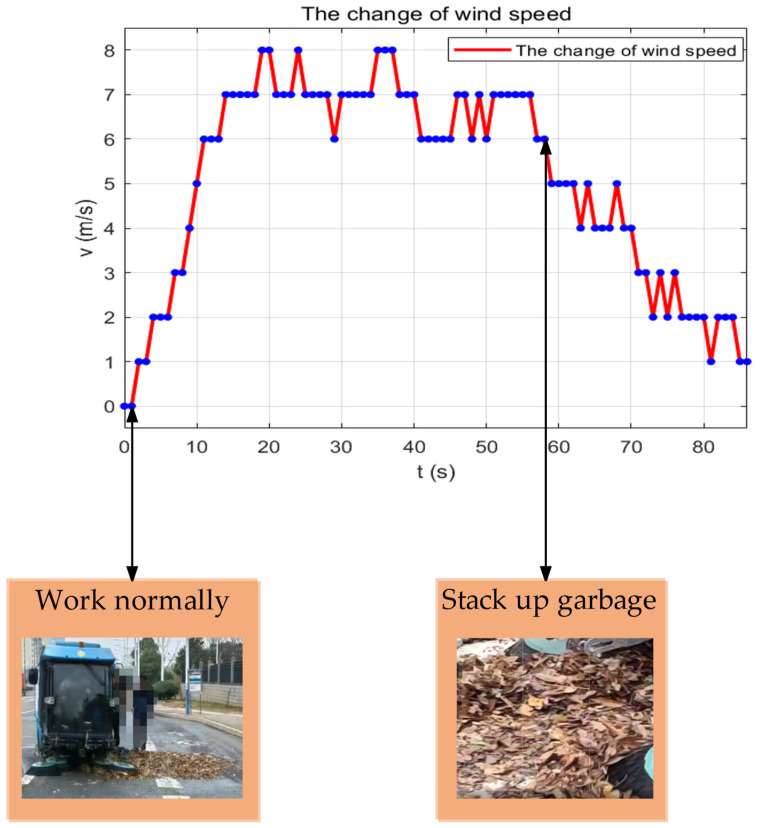
Air velocity fluctuations in ventilation ducts.

**Table 1 sensors-25-04010-t001:** Comparison of the cascade algorithm’s performance with that of a single or dual module.

Algorithmic Module	Noise Standard Deviation (mm)	Impulse Noise Rejection (%)
raw data	18.7	0.0
IQR	13.1	68.2
SG	4.2	23.5
Kalman	2.3	41.7
IQR + SG	7.5	72.8
SG + Kalman	3.6	58.3
IQR + Kalman	4.8	89.1
IQR → SG → Kalman	2.1	98.5

## Data Availability

Data are contained within the article.

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
