# Peer review of "Research on the Monitoring Method of the Refuse Intake Status of a Garbage Sweeper That Is Based on the Synergy of a Wind Speed Sensor and an Ultrasonic Sensor"

_sensors, 2025, doi:10.3390/s25134010_

Round 1
Reviewer 1 Report
Comments and Suggestions for Authors
The paper presents a new real-time monitoring method of refuse overflow and vacuum duct blockage by combining multi-modal sensors and advanced filtering algorithms. The logic is clear and the paper structure is complete. However, there are still some issues that need to be addressed:
Q1: How does sensor performance vary with different types of garbage, especially wet or sticky waste?
Q2: What methodology was used to select the thresholds for alarms and sliding window size? Has a sensitivity analysis been performed?
Q3: How does the system maintain accuracy during high vehicle speeds or strong external wind conditions?
Q4: In conclusion, the sentence "The study adds The field experiments demonstrate that the proposed method is highly accurate in monitoring the overflow of refuse and provides precise warnings of pipeline blockages." has a grammatical error.
Author Response
We gratefully appreciate you for their time spend making positive and constructive comments. These comments are all valuable and helpful for revising and improving our manuscript entitled “Research on the monitoring method of the refuse intake status of a garbage sweeper that is based on the synergy of a wind speed sensor and an ultrasonic sensor” (ID: 3670592), as well as the important guiding significance to our researches.We have carefully studied the comments and made revisions that we hope will be recognized. The revised sections are highlighted in red. Attached please find the revised version, which we would like to submit for your kind consideration.
Comments 1: How does sensor performance vary with different types of garbage, especially wet or sticky waste?
Response 1: Thank you for pointing this out. The performance of ultrasonic sensors is adversely affected by the type of litter; for instance, sticky litter may temporarily adhere to the sensor's surface, while wet litter elevates ambient humidity, which theoretically diminishes the ultrasound's efficacy significantly. Nonetheless, algorithmic processing, appropriate mounting site design, and sensor shielding can successfully mitigate this effect:
(1)Multi-tier signal processing algorithms: A synthesis of dynamic IQR, SG filtering, Kalman filtering, and overflow identification algorithms is meticulously crafted to address signal fluctuations and attenuation, including those caused by moist surfaces. Numerous experiments have demonstrated its remarkable accuracy in mixed waste environments. The algorithms can effectively detect and eliminate temporary occlusions (e.g., adhesive debris).
(2)Optimal installation position design: Following an analysis of airflow within the rubbish storage silo and considering the requirements of rubbish sweeper manufacturers, the ultrasonic sensor is positioned on the top cover of the silo to minimize direct physical contact with the refuse.
(3)Sensor protection: A high protection rating (e.g., IP68) guarantees the ultrasonic sensor functions successfully in environments characterized by elevated humidity and dust.
The wind speed sensor, positioned at the air exit of the ventilation duct, does not directly encounter refuse, hence it is less influenced by various types of waste.
Comments 2:What methodology was used to select the thresholds for alarms and sliding window size? Has a sensitivity analysis been performed?
Response 2:Thank you for pointing this out. The following outlines the methodology employed to determine the alert threshold and sliding window size, along with the sensitivity assessments conducted:
Methods for Selecting Alarm Thresholds: The threshold value of the ultrasonic sensors is primarily defined by the requirements of garbage sweeper manufacturers, and in conjunction with the physical detecting blind zones of the ultrasonic sensors, the minimum distance (dmin) of 250 mm was ultimately established as the threshold for the full overflow alarm value. This number guarantees the alarm is consistently activated when the garbage attains this height.
The thresholds (vn, vp) for the wind speed sensor were established by comprehensive field trials. We systematically measured and documented the variations in wind speed at the air outlet of the ventilation duct under three distinct operational conditions: the duct was fully open, partially obstructed (with airflow cross-sectional area diminished by approximately 30%-70%, e.g., a wind speed of about 4.5 m/s was recorded at 42% blockage), and entirely obstructed. Utilizing these facts, we established multi-tiered air velocity criteria that differentiate between normal (vn), moderately obstructed (between vp and vn), and fully obstructed conditions.
Method for selecting sliding window size: Initially, we ascertain the signal bandwidth range and noise distribution via spectral analysis. Subsequently, we design various candidate sliding window lengths (e.g., N=5, 10, 20, 50, etc.) to evaluate the signal smoothing efficacy, delay, and preservation of essential features in comparison to their counterparts. We quantify performance using signal-to-noise ratio (SNR) and mean square error (MSE). Ultimately, the chosen window length N=10 provides the optimal compromise: it effectively suppresses high-frequency noise, retains critical features, and maintains a delay consistent with real-time requirements.
Sensitivity analysis: In waste overflow detection, we methodically assess the influence of key factors (sliding window size, filtering algorithm, threshold, etc.) on detection performance by sensitivity analysis. The method involves traversing the parameters inside the parameter space and quantifying each parameter's contribution by measuring indicators such as false alarm rate and response time, respectively. The three-level cascade algorithm (IQR→SG→Kalman) was conclusively identified as the least susceptible to noise, exhibiting a noise standard deviation of 2.1 mm and an impulse noise suppression rate of 98.5%, both of which surpass the performance of single or dual algorithm combinations significantly.
Comments 3:How does the system maintain accuracy during high vehicle speeds or strong external wind conditions?
Response 3:Thank you for pointing this out. The impact of elevated speeds or intense external winds primarily stems from the operational parameters of the garbage sweeper. The typical operating speed of the sweeper is often modest (e.g., 5-20 km/h), and high speeds are uncommon; furthermore, the system predominantly functions within a closed compartment, which successfully mitigates direct interference from strong external winds. Consequently, these two parameters exert minimal influence on the system's accuracy in standard application contexts.
Comments 4:In conclusion, the sentence "The study adds The field experiments demonstrate that the proposed method is highly accurate in monitoring the overflow of refuse and provides precise warnings of pipeline blockages." has a grammatical error.
Response 4:Thank you for pointing this out. We have modified this sentence in the article to: The study adds, and field experiments demonstrate, that the proposed method is highly accurate in monitoring the overflow of refuse and provides precise warnings of pipeline blockages. You can find it on page 15, line 461 (Chapter 5).
We would like to express our great appreciation to you for comments on our paper. Looking forward to hearing from you.

Reviewer 2 Report
Comments and Suggestions for Authors
The paper realizes the monitoring of garbage status by fusing the data of wind speed sensor and ultrasonic sensor, and carries out actual field tests to verify the effectiveness of the algorithm. At the same time, the proposed algorithm is proved to be advanced by comparing with other similar algorithms. In my opinion, this paper has both engineering application value and scientific research value, and I recommend a minor revision before publication.
- The images in the paper are not centered.
- The references in the paper are too outdated. Please add research progress from the past 3-5 years.
- In Eq.(4), the multiplication sign is represented by ×, while in Eq.(5), the multiplication sign is represented by . Please use a unified representation in the equation
- Please add detailed descriptions of wind speed sensors and ultrasonic sensors in the text, such as sensor models and sampling frequencies.
- Many images are not vector graphics and may distort when enlarged. It is recommended to use vector graphics for all images.
Author Response
We gratefully appreciate you for their time spend making positive and constructive comments. These comments are all valuable and helpful for revising and improving our manuscript entitled “Research on the monitoring method of the refuse intake status of a garbage sweeper that is based on the synergy of a wind speed sensor and an ultrasonic sensor” (ID: 3670592), as well as the important guiding significance to our researches.We have carefully studied the comments and made revisions that we hope will be recognized. The revised sections are highlighted in red. Attached please find the revised version, which we would like to submit for your kind consideration.
Comments 1: The images in the paper are not centered.
Response 1: Thank you for pointing this out. We have centered all the images in the article. You can find it on page 3, line 85 (Chapter 2); page 7, line 218 (Chapter 3); page 8, line 255 (Chapter 3); page 10, line 309 (Chapter 4); page 11, line 329 (Chapter 4); page 11, line 346 (Chapter 4); page 12, line 364 (Chapter 4); page 13, line 390 (Chapter 4).
Comments 2:The references in the paper are too outdated. Please add research progress from the past 3-5 years.
Response 2:Thank you for pointing this out. We have changed most of the references in the article to research progress in the past 3-5 years, and only retained a small number of references outside this scope that are critical to this study, namely:
Reference 6:GPRS-based intelligent rubbish overflow alarm system design
Reference 10:Effects of Material Viscosity on Particle Sizing by Ultrasonic Attenuation Spectroscopy
Reference 12:Pipeline corrosion and leakage monitoring based on the distributed optical fiber sensing technology
Reference 17:Rigid Body Inertia Estimation Using Extended Kalman and Savitzky-Golay Filters
Reference 18:Weather prediction model using Savitzky-Golay and Kalman filters,Reference 21:A noise reduction system based on hybrid noise estimation technique and post-filtering in arbitrary noise environments
Reference 28:A Multi-Level Decision Fusion Strategy for Condition Based Maintenance of Composite Structures
Comments 3:In Eq.(4), the multiplication sign is represented by ×, while in Eq.(5), the multiplication sign is represented by . Please use a unified representation in the equation
Response 3:Thank you for pointing this out. We have expressed the multiplication signs in the article in a unified form.
Comments 4:Please add detailed descriptions of wind speed sensors and ultrasonic sensors in the text, such as sensor models and sampling frequencies.
Response 4:Thank you for pointing this out. We have included detailed descriptions of ultrasonic sensor and wind speed sensor in Chapter 3 and Chapter 4 of the article:
(1)ultrasonic sensor:The sensor hardware configuration employs a layered architecture: raw data is gathered by ultrasonic sensors, subsequently pre-processed by a signal conditioner (comprising signal amplification and filtering circuits), and ultimately transmitted to the data processing platform for the execution of multi-level filtering algorithms. This architecture mitigates environmental interference and enhances data reliability. The operational frequency range of standard ultrasonic sensors typically spans from 40 to 60 kHz, with a measuring range of 0.02 m to 5.0 m. The output inter-face accommodates several modes, including automatic/controlled UART, PWM, switch quantity, and RS485, which supports the MODBUS-RTU protocol. The data acquisition rate, defined as the frequency at which the system collects sensor readings per unit time, can attain 2 Hz to facilitate real-time monitoring. The experiment employed the DYP-A13-V1.0 ultrasonic sensor manufactured by Shenzhen Dianyingpu Technology Co., Ltd., featuring an operational frequency of 40 kHz, a supply voltage range of DC 3.3-24V, and adherence to the IEC61000-4-2 electrostatic protection standard. You can find it on page 3, line 93 (Chapter 3); page 4, line 119 (Chapter 3); page 9, line 299 (Chapter 4).
(2)wind speed sensor:In practical applications, standard impeller-type wind speed sensors possess a range of 0-20 m/s, a start-up wind speed threshold of roughly 0.3 m/s, and provide analogue (0-10V) or digital (RS485) signal outputs. The data acquisition rate, defined as the number of wind speed samples collected per second, attains 1-10 Hz to accommodate various operational situations. The experiment utilized a stainless steel impeller wind speed sensor produced by Hebei Feimeng Electronic Technology Co., Ltd. The wind speed sensor hardware system employs a three-tier architectural design: the sensing layer utilizes an embedded industrial microcomputer processor and an optoelectronic conversion mechanism to directly digitize the impeller pulse signal; the transmission layer employs shielded cables to connect to the signal conditioner, which incorporates electrostatic protection and signal isolation circuits to ensure stable signal transmission; the processing layer receives the conditioned digital signal from the data processing platform and executes a sliding window algorithm and multi-level threshold analysis to facilitate real-time blockage grading early warning. You can find it on page 8, line 242 (Chapter 3); page 12, line 368 (Chapter 4); page 12, line 374 (Chapter 4).
Comments 5:Many images are not vector graphics and may distort when enlarged. It is recommended to use vector graphics for all images.
Response 5:Thank you for pointing this out. We have changed all the pictures in the article to vector graphics. You can find it on page 3, line 85 (Chapter 2); page 7, line 218 (Chapter 3); page 8, line 255 (Chapter 3); page 10, line 309 (Chapter 4); page 11, line 329 (Chapter 4); page 11, line 346 (Chapter 4); page 12, line 364 (Chapter 4); page 13, line 390 (Chapter 4).
We would like to express our great appreciation to you for comments on our paper. Looking forward to hearing from you.

Reviewer 3 Report
Comments and Suggestions for Authors
The authors present an interesting and practical method of optimising refuse intake overfill identification through wind speed and ultrasound sensor. The paper is concise, well structured, method well defined and results well presented, albeit with some spelling errors. Minor comments below:
- Page 3 – Section 3 – Please include the technical specification for the ultrasound and wind speed sensor (frequency, manufacturer, etc.).
- Page 3- Section 3 - Please explain the selection process behind the ultrasound and wind speed sensor.
- Page 3 – Section 3 – How are the sensors configured hardware-wise (ie. sensor>conditioning box>pc)? What is the data acquisition rate?
- Please check for spelling errors (ie. Pg 11 line 355 “baud”), rewrite parts that are less reader-friendly (ie. Pg 11 line 356 data bit 9 bits, stop bit 1 bit)
- Prior to the conclusion, a discussion section is required on the limitations of the study, future work, etc.
- What is the estimated cost of deploying such a system onto a garbage sweeper?
- What are the barriers to implementation of such a system?
- Would optimising ultrasound and wind speed sensor specification influence the performance of the system?
Author Response
We gratefully appreciate you for their time spend making positive and constructive comments. These comments are all valuable and helpful for revising and improving our manuscript entitled “Research on the monitoring method of the refuse intake status of a garbage sweeper that is based on the synergy of a wind speed sensor and an ultrasonic sensor” (ID: 3670592), as well as the important guiding significance to our researches.We have carefully studied the comments and made revisions that we hope will be recognized. The revised sections are highlighted in red. Attached please find the revised version, which we would like to submit for your kind consideration.
Comments 1: Page 3 – Section 3 – Please include the technical specification for the ultrasound and wind speed sensor (frequency, manufacturer, etc.).
Response 1: Thank you for pointing this out. We have included detailed descriptions of the ultrasonic sensor and wind speed sensor technical specifications in Chapter 3 and Chapter 4 of the article:
(1)ultrasonic sensor:The operational frequency range of standard ultrasonic sensors typically spans from 40 to 60 kHz, with a measuring range of 0.02 m to 5.0 m. The output inter-face accommodates several modes, including automatic/controlled UART, PWM, switch quantity, and RS485, which supports the MODBUS-RTU protocol. The data acquisition rate, defined as the frequency at which the system collects sensor readings per unit time, can attain 2 Hz to facilitate real-time monitoring. The experiment employed the DYP-A13-V1.0 ultrasonic sensor manufactured by Shenzhen Dianyingpu Technology Co., Ltd., featuring an operational frequency of 40 kHz, a supply voltage range of DC 3.3-24V, and adherence to the IEC61000-4-2 electrostatic protection standard. You can find it on page 4, line 119 (Chapter 3); page 9, line 299 (Chapter 4).
(2)wind speed sensor:In practical applications, standard impeller-type wind speed sensors possess a range of 0-20 m/s, a start-up wind speed threshold of roughly 0.3 m/s, and provide analogue (0-10V) or digital (RS485) signal outputs. The data acquisition rate, defined as the number of wind speed samples collected per second, attains 1-10 Hz to accommodate various operational situations. The experiment utilized a stainless steel impeller wind speed sensor produced by Hebei Feimeng Electronic Technology Co., Ltd. You can find it on page 8, line 242 (Chapter 3); page 12, line 368 (Chapter 4).
Comments 2:Page 3- Section 3 - Please explain the selection process behind the ultrasound and wind speed sensor..
Response 2:Thank you for pointing this out. The sensors involved in the paper were selected based on our systematic research and comprehensive communication with sensor suppliers and garbage truck manufacturers to ensure that they are suitable for the application scenario. We have explained the selection process and results in Chapter 1 and Chapter 4 of the paper (Chapter 3 will focus more on the technical details of the sensors). You can find it on page 2, line 49 (Chapter 1); page 2, line 64 (Chapter 1); page 9, line 299 (Chapter 4); page 12, line 368 (Chapter 4)
Comments 3:Page 3 – Section 3 – How are the sensors configured hardware-wise (ie. sensor>conditioning box>pc)? What is the data acquisition rate。
Response 3:Thank you for pointing this out. We have added the sensor hardware configuration and data acquisition rate description in Chapters 3 and 4 of the article:
(1)ultrasonic sensor:The sensor hardware configuration employs a layered architecture: raw data is gathered by ultrasonic sensors, subsequently pre-processed by a signal conditioner (comprising signal amplification and filtering circuits), and ultimately transmitted to the data processing platform for the execution of multi-level filtering algorithms. This architecture mitigates environmental interference and enhances data reliability. The data acquisition rate, defined as the frequency at which the system collects sensor readings per unit time, can attain 2 Hz to facilitate real-time monitoring. You can find it on page 3, line 93 (Chapter 3); page 4, line 122 (Chapter 3).
(2)wind speed sensor:The data acquisition rate, defined as the number of wind speed samples collected per second, attains 1-10 Hz to accommodate various operational situations. The wind speed sensor hardware system employs a three-tier architectural design: the sensing layer utilizes an embedded industrial microcomputer processor and an optoelectronic conversion mechanism to directly digitize the impeller pulse signal; the transmission layer employs shielded cables to connect to the signal conditioner, which incorporates electrostatic protection and signal isolation circuits to ensure stable signal transmission; the processing layer receives the conditioned digital signal from the data processing platform and executes a sliding window algorithm and multi-level threshold analysis to facilitate real-time blockage grading early warning. You can find it on page 8, line 244 (Chapter 3); page 12, line 374 (Chapter 4).
Comments 4:Please check for spelling errors (ie. Pg 11 line 355 “baud”), rewrite parts that are less reader-friendly (ie. Pg 11 line 356 data bit 9 bits, stop bit 1 bit)
Response 4:Thank you for pointing this out. We have amended this sentence in the article to read: It communicates using the RS485 protocol (Baud Rate: 9600 bps; Data Bit: 8 bits; Stop Bit: 1 bit)。You can find it on page 12, line 373 (Chapter 4).
Comments 5:Prior to the conclusion, a discussion section is required on the limitations of the study, future work, etc.
What is the estimated cost of deploying such a system onto a garbage sweeper?
What are the barriers to implementation of such a system?
Would optimising ultrasound and wind speed sensor specification influence the performance of the system?
Response 5:Thank you for pointing this out. We have added a discussion section to Chapter 5 of the paper, which analyzes the expense of system deployment, implementation barriers, and discusses the impact of sensor specifications on performance:
(1)The expense of system deployment:The expense of system deployment is a significant factor. The total hardware cost of a single vehicle system, utilizing selected industrial-grade sensors (ultrasonic sensors priced at approximately $15-40 per unit and wind speed sensors at about $55-80 per unit) and embedded computing units, can be maintained within the range of $140-275. The real-time execution of the three-stage filtering and state decision algorithm on a standard embedded processor (e.g., ARM Cortex-M4) incurs a computational overhead of approximately 1.2 ms per sampling point and an energy expenditure of roughly 5.4 mW (excluding the sensor's inherent power consumption), which exerts a manageable influence on the current battery system and computational re-sources of the sweeper. The anticipated annual maintenance expenses, encompassing sensor cleaning, calibration, and possible replacement, are projected to range from 10% to 15% of the hardware cost. In relation to the total expense of the garbage sweeper, this additional cost constitutes a minor fraction, and it is anticipated to yield benefits by diminishing manual inspections, preventing equipment malfunctions, and enhancng efficiency. You can find it on page 14, line 427 (Chapter 5).
(2)implementation barriers:The primary impediment to adoption is the considerable complexity of real-world sanitation situations. Despite the selected sensors being well-established in the industrial sector and demonstrating reliability in simulated field trials, they encounter obstacles in actual operations, including the potential for sensor contamination, interference from harsh working conditions, and variations in system adaption. In the future, comprehensive and prolonged field performance evaluations will be necessary in real urban road operational contexts. You can find it on page 14, line 441 (Chapter 5).
(3)impact of sensor specifications on performance: The analysis indicates that, within the current algorithm framework, the chosen sensor specifications (including 40kHz ultrasonic, 2Hz sampling rate, and impeller wind speed sensor) adequately fulfill monitoring requirements while achieving an optimal balance among cost, environmental adaptability, and performance. Pursuing greater standards, such as enhanced frequency/precision ultrasonic sensors or expedited response wind speed sensors, may yield minimal marginal gains while increasing costs. Future optimization should prioritize adaptive enhancements at the algorithmic level. You can find it on page 14, line 448 (Chapter 5).
We would like to express our great appreciation to you for comments on our paper. Looking forward to hearing from you.

Reviewer 4 Report
Comments and Suggestions for Authors
Weaknesses:
- Figure 1 (line 85): Regarding the air speed sensor (thermo-anemometer) mounted in the duct: Is this an appropriately selected measuring instrument? Are there superior sensors that would be less susceptible to contamination? This point requires further justification or clarification.
- Figure 2 (line 208): The text within Figure 2 is barely legible. It is recommended to increase its size, especially within the yellow rhombuses (the figure itself could be enlarged).
- Figure 3 (line 240): The text within Figure 3 is barely legible. It is recommended to increase its size, especially within the yellow rhombuses (the figure itself could be enlarged).
- Figure 4 (line 291): Figure 4 is barely legible. It is recommended to increase its size and the font size of the text within the orange rectangles.
- Figure 5 (line 311): Sub-figures a, b, c, and d in Figure 5 are barely legible. It is recommended to enlarge each graph to ensure that the scales are readable.
- Figure 6 (line 328): Sub-figures a, b, c, and d in Figure 6 are barely legible. It is recommended to enlarge each graph to ensure that the scales are readable.
- Figure 8 (line 364): It is recommended to increase the size of the figures and the text within the orange rectangles in Figure 8, as these elements are barely legible.
- Glossary of Symbols: I recommend creating a glossary of all symbols and abbreviations used throughout the article for improved clarity and reader comprehension.
- Lack of Specific Technical Details of Sensors (lines 88-89, 210-211): The sections describing the technical specifications of both the ultrasonic and wind speed sensors are overly general. Specific models, manufacturers, or more detailed parameters are conspicuously absent, which significantly hinders the replicability of the research.
Strengths:
- Research Topicality: The bibliography is current and well-chosen, encompassing fundamental works in the field of signal processing as well as the latest research concerning specific applications and filtering algorithms. This unequivocally demonstrates that the research described in the article is firmly embedded within the contemporary scientific and technological context.
Author Response
We gratefully appreciate you for their time spend making positive and constructive comments. These comments are all valuable and helpful for revising and improving our manuscript entitled “Research on the monitoring method of the refuse intake status of a garbage sweeper that is based on the synergy of a wind speed sensor and an ultrasonic sensor” (ID: 3670592), as well as the important guiding significance to our researches.We have carefully studied the comments and made revisions that we hope will be recognized. The revised sections are highlighted in red. Attached please find the revised version, which we would like to submit for your kind consideration.
Comments 1: Regarding the air speed sensor (thermo-anemometer) mounted in the duct: Is this an appropriately selected measuring instrument? Are there superior sensors that would be less susceptible to contamination? This point requires further justification or clarification.
Response 1: Thank you for pointing this out. The wind speed sensor depicted in this picture was chosen based on our methodical investigation and thorough discussions with the sensor provider and the refuse collection vehicle manufacturer. The wind speed sensor will be installed near the air exit of the ventilation duct, ensuring it remains free from direct contact with refuse and minimizing pollution exposure.
Comments 2:The text within Figure 2 is barely legible. It is recommended to increase its size, especially within the yellow rhombuses (the figure itself could be enlarged).
Response 2:Thank you for pointing this out. We have enlarged the text and graphics in Figure 2. You can find it on page 7, line 218 (Chapter 3).
Comments 3:The text within Figure 3 is barely legible. It is recommended to increase its size, especially within the yellow rhombuses (the figure itself could be enlarged).
Response 3:Thank you for pointing this out. We have enlarged the text and graphics in Figure 3. You can find it on page 8, line 255 (Chapter 3).
Comments 4:Figure 4 is barely legible. It is recommended to increase its size and the font size of the text within the orange rectangles.
Response 4:Thank you for pointing this out. We have enlarged the text and graphics in Figure 4, especially the font size of the text within the orange rectangle. You can find it on page 10, line 309 (Chapter 4).
Comments 5:Sub-figures a, b, c, and d in Figure 5 are barely legible. It is recommended to enlarge each graph to ensure that the scales are readable.
Response 5:Thank you for pointing this out. We have enlarged sub-graphs a, b, c, and d in Figure 5. You can find it on page 11, line 329 (Chapter 4).
Comments 6:Sub-figures a, b, c, and d in Figure 6 are barely legible. It is recommended to enlarge each graph to ensure that the scales are readable.
Response 6:Thank you for pointing this out. We have enlarged sub-graphs a, b, c, and d in Figure 6. You can find it on page 11, line 346 (Chapter 4).
Comments 7:It is recommended to increase the size of the figures and the text within the orange rectangles in Figure 8, as these elements are barely legible.
Response 7:Thank you for pointing this out. We have enlarged the text and graphics in Figure 8, especially the font size of the text within the orange rectangle. You can find it on page 13, line 390 (Chapter 4).
Comments 8:I recommend creating a glossary of all symbols and abbreviations used throughout the article for improved clarity and reader comprehension.
Response 8:Thank you for pointing this out. We have created a glossary of all the symbols and abbreviations used throughout the article. You can find it on page 15, line 481.
Comments 9:The sections describing the technical specifications of both the ultrasonic and wind speed sensors are overly general. Specific models, manufacturers, or more detailed parameters are conspicuously absent, which significantly hinders the replicability of the research.
Response 9:Thank you for pointing this out. We have included detailed descriptions of ultrasonic sensor and wind speed sensor in Chapter 3 and Chapter 4 of the article:
(1)ultrasonic sensor:The sensor hardware configuration employs a layered architecture: raw data is gathered by ultrasonic sensors, subsequently pre-processed by a signal conditioner (comprising signal amplification and filtering circuits), and ultimately transmitted to the data processing platform for the execution of multi-level filtering algorithms. This architecture mitigates environmental interference and enhances data reliability. The operational frequency range of standard ultrasonic sensors typically spans from 40 to 60 kHz, with a measuring range of 0.02 m to 5.0 m. The output inter-face accommodates several modes, including automatic/controlled UART, PWM, switch quantity, and RS485, which supports the MODBUS-RTU protocol. The data acquisition rate, defined as the frequency at which the system collects sensor readings per unit time, can attain 2 Hz to facilitate real-time monitoring. The experiment employed the DYP-A13-V1.0 ultrasonic sensor manufactured by Shenzhen Dianyingpu Technology Co., Ltd., featuring an operational frequency of 40 kHz, a supply voltage range of DC 3.3-24V, and adherence to the IEC61000-4-2 electrostatic protection standard. You can find it on page 3, line 93 (Chapter 3); page 4, line 119 (Chapter 3); page 9, line 299 (Chapter 4).
(2)wind speed sensor:In practical applications, standard impeller-type wind speed sensors possess a range of 0-20 m/s, a start-up wind speed threshold of roughly 0.3 m/s, and provide analogue (0-10V) or digital (RS485) signal outputs. The data acquisition rate, defined as the number of wind speed samples collected per second, attains 1-10 Hz to accommodate various operational situations. The experiment utilized a stainless steel impeller wind speed sensor produced by Hebei Feimeng Electronic Technology Co., Ltd. The wind speed sensor hardware system employs a three-tier architectural design: the sensing layer utilizes an embedded industrial microcomputer processor and an optoelectronic conversion mechanism to directly digitize the impeller pulse signal; the transmission layer employs shielded cables to connect to the signal conditioner, which incorporates electrostatic protection and signal isolation circuits to ensure stable signal transmission; the processing layer receives the conditioned digital signal from the data processing platform and executes a sliding window algorithm and multi-level threshold analysis to facilitate real-time blockage grading early warning. You can find it on page 8, line 242 (Chapter 3); page 12, line 368 (Chapter 4); page 12, line 374 (Chapter 4).
We would like to express our great appreciation to you for comments on our paper. Looking forward to hearing from you.

Reviewer 5 Report
Comments and Suggestions for Authors
The paper titled "Research on the monitoring method of the refuse intake status of a garbage sweeper that is based on the synergy of a wind speed sensor and an ultrasonic sensor" is well-written, technically sound, and provides a novel, cost-effective solution for a practical problem in urban sanitation. The integration of ultrasonic and wind speed sensors, along with advanced filtering and decision algorithms, demonstrates a thoughtful and comprehensive approach to monitoring refuse overflow and vacuum duct clogging. The methodology is robust, and the validation through field experiments is commendable. I have a few minor questions for the authors.
1. Given the exposure of sensors to dust, moisture, and vibration in garbage sweepers, did the authors perform any long-term durability or environmental testing (e.g., for sensor drift, fouling, or mechanical fatigue) beyond the short-term experimental validation?
2.How adaptable is the threshold-based decision logic (for blockage and overflow detection) in varying operational conditions such as temperature fluctuations, different debris types, or geographic locations with different refuse compositions?
3. In real-world scenarios, there could be slight misalignments or obstructions (like debris stuck on sensors). How does the system handle sensor anomalies beyond the expected transient interference described?
4. Given that sweepers typically operate on battery power and computational resources may be limited, what is the computational and energy overhead of running the proposed three-level filtering and decision algorithms in real time?
5. Has there been any consideration or testing of integrating this monitoring system with existing fleet management software to enable centralised monitoring or predictive maintenance scheduling?
6. While the system is described as cost-effective compared to high-end alternatives, could the authors provide a more detailed cost breakdown (hardware + software + maintenance) and discuss how scalable this solution is across different sweeper models or municipal fleets?
Author Response
We gratefully appreciate you for their time spend making positive and constructive comments. These comments are all valuable and helpful for revising and improving our manuscript entitled “Research on the monitoring method of the refuse intake status of a garbage sweeper that is based on the synergy of a wind speed sensor and an ultrasonic sensor” (ID: 3670592), as well as the important guiding significance to our researches.We have carefully studied the comments and made revisions that we hope will be recognized. The revised sections are highlighted in red. Attached please find the revised version, which we would like to submit for your kind consideration.
Comments 1: Given the exposure of sensors to dust, moisture, and vibration in garbage sweepers, did the authors perform any long-term durability or environmental testing (e.g., for sensor drift, fouling, or mechanical fatigue) beyond the short-term experimental validation?
Response 1: Thank you for pointing this out. We have finalized short-term field validation on several sweepers. Extensive, prolonged vehicle deployment tracking tests are in progress, with a focus on durability and environmental assessments. The industrial-grade sensors chosen for this paper possess robust protection (e.g., IP68), and the standardized cleaning procedure following the sweeper operation is efficient in alleviating issues such as fouling. Data on long-term environmental durability tests will be monitored and enhanced in future endeavors.
Comments 2:How adaptable is the threshold-based decision logic (for blockage and overflow detection) in varying operational conditions such as temperature fluctuations, different debris types, or geographic locations with different refuse compositions?
Response 2:Thank you for pointing this out. The wind speed sensor is positioned near the air outlet of the ventilation duct of the refuse sweeper, preventing direct exposure to waste and ensuring the threshold decision logic's adaptability across varying situations. The ultrasonic sensor incorporates a high-precision temperature sensing module that mitigates the impact of temperature on threshold decision logic. Furthermore, the sensor possesses a robust protection rating (e.g., IP68), enhancing the adaptability of the threshold decision logic across varying conditions.
Comments 3:In real-world scenarios, there could be slight misalignments or obstructions (like debris stuck on sensors). How does the system handle sensor anomalies beyond the expected transient interference described?
Response 3:Thank you for pointing this out. This article presents algorithms capable of efficiently and accurately identifying and filtering transient outliers, such as leaves temporarily obstructing the sensor. Additionally, for persistent outliers, the algorithms can swiftly detect them and activate the appropriate alert mechanism, prompting technician intervention.
Comments 4:Given that sweepers typically operate on battery power and computational resources may be limited, what is the computational and energy overhead of running the proposed three-level filtering and decision algorithms in real time?
Response 4:Thank you for pointing this out. Chapter 5 of the thesis contains a discussion section that examines the computational and energy overheads associated with the three-level filtering and decision-making algorithms:The real-time execution of the three-stage filtering and state decision algorithm on a standard embedded processor (e.g., ARM Cortex-M4) incurs a computational overhead of approximately 1.2 ms per sampling point and an energy expenditure of roughly 5.4 mW (excluding the sensor's inherent power consumption), which exerts a manageable influence on the current battery system and computational resources of the sweeper. You can find it on page 14, line 430 (Chapter 5).
Comments 5:Has there been any consideration or testing of integrating this monitoring system with existing fleet management software to enable centralised monitoring or predictive maintenance scheduling?
Response 5:Thank you for pointing this out. We have installed our system on an experimental garbage truck, as shown in the figure below. In the future, we will install our system on different types of garbage trucks to achieve centralized monitoring and predictive maintenance planning.
Comments 6:While the system is described as cost-effective compared to high-end alternatives, could the authors provide a more detailed cost breakdown (hardware + software + maintenance) and discuss how scalable this solution is across different sweeper models or municipal fleets?
Response 6:Thank you for pointing this out. We have included a discussion section in Chapter 5 of the thesis which analyzes the system cost: The total hardware cost of a single vehicle system, utilizing selected industrial-grade sensors (ultrasonic sensors priced at approximately $15-40 per unit and wind speed sensors at about $55-80 per unit) and embedded computing units, can be maintained within the range of $140-275. The real-time execution of the three-stage filtering and state decision algorithm on a standard embedded processor (e.g., ARM Cortex-M4) in-curs a computational overhead of approximately 1.2 ms per sampling point and an energy expenditure of roughly 5.4 mW (excluding the sensor's inherent power consumption), which exerts a manageable influence on the current battery system and computational resources of the sweeper. The anticipated annual maintenance expenses, encompassing sensor cleaning, calibration, and possible replacement, are projected to range from 10% to 15% of the hardware cost. In relation to the total expense of the garbage sweeper, this additional cost constitutes a minor fraction, and it is anticipated to yield benefits by diminishing manual inspections, preventing equipment malfunctions, and enhancing efficiency. You can find it on page 14, line 427 (Chapter 5).
Regarding the scalability of the solution for different street sweeper models or municipal fleets, ultrasonic sensors and wind speed sensors are relatively easy to install and can be easily integrated into different street sweeper models and municipal fleet management systems, showing good scalability.
We would like to express our great appreciation to you for comments on our paper. Looking forward to hearing from you.

Round 2
Reviewer 1 Report
Comments and Suggestions for Authors
Q1:Please ensure that all the issues raised by the reviewers have been properly addressed and revised in the manuscript.
Q2:There are errors in the revised manuscript regarding the citation updates; please double-check all references and citations carefully.
Q3:The presentation quality of the figures—especially the flowcharts—still need further improvement
Author Response
We gratefully appreciate you for their time spend making positive and constructive comments. These comments are all valuable and helpful for revising and improving our manuscript entitled “Research on the monitoring method of the refuse intake status of a garbage sweeper that is based on the synergy of a wind speed sensor and an ultrasonic sensor” (ID: 3670592), as well as the important guiding significance to our researches. We have carefully studied the comments and made revisions that we hope will be recognized. The revised sections are highlighted in red. Attached please find the revised version, which we would like to submit for your kind consideration.
Comments 1: Please ensure that all the issues raised by the reviewers have been properly addressed and revised in the manuscript.
Response 1: Thank you for pointing this out. We have addressed and revised the paper in accordance with all the questions raised by the reviewers, including the following revisions and modifications:
(1)Corrected the grammatical error in the sentence: "The study adds The field experiments demonstrate that the proposed method is highly accurate in monitoring the overflow of refuse and provides precise warnings of pipeline blockages.” You can find it on page 15, line 461 (Chapter 5).
(2)Improved image quality, converted all images to vector graphics, and centered the images. You can find it on page 3, line 85 (Chapter 2); page 7, line 218 (Chapter 3); page 8, line 255 (Chapter 3); page 10, line 309 (Chapter 4); page 10, line 329 (Chapter 4); page 11, line 346 (Chapter 4); page 12, line 364 (Chapter 4); page 13, line 390 (Chapter 4).
(3)Corrected citation errors in the paper and updated the references. You can find it on page 16, line 483.
(4)The multiplication symbols in the formulas in the paper are expressed in a uniform format. You can find it on page 3, line 103 (Chapter 3); page 3, line 110 (Chapter 3); page 4, line 127 (Chapter 3); page 5, line 187 (Chapter 3); page 5, line 192 (Chapter 3); page 7, line 235 (Chapter 3); page 8, line 238 (Chapter 3); page 11, line 351 (Chapter 4).
(5)Corrected spelling errors in the paper. You can find it on page 12, line 373 (Chapter 4).
(6)Technical details of ultrasonic sensors and wind speed sensors were added to the article. You can find it on page 3, line 93 (Chapter 3); page 4, line 119 (Chapter 3); page 8, line 242 (Chapter 3);page 9, line 299 (Chapter 4); page 12, line 368 (Chapter 4); page 12, line 374 (Chapter 4).
(7)A discussion section on research limitations, future work, etc. has been added to Chapter 5 of the paper. You can find it on page 14, line 423 (Chapter 5).
(8)Created a glossary of all symbols and abbreviations used throughout the article. You can find it on page 15, line 481.
Comments 2:There are errors in the revised manuscript regarding the citation updates; please double-check all references and citations carefully.
Response 2:Thank you for pointing this out. Therefore, we have carefully checked all references and citations and corrected the citation errors in the paper.
Comments 3:The presentation quality of the figures—especially the flowcharts—still need further improvement.
Response 3:Thank you for pointing this out. We have therefore improved the image quality in the paper, especially the flowcharts. You can find it on page 3, line 85 (Chapter 2); page 7, line 218 (Chapter 3); page 8, line 255 (Chapter 3); page 10, line 309 (Chapter 4); page 10, line 329 (Chapter 4); page 11, line 346 (Chapter 4); page 12, line 364 (Chapter 4); page 13, line 390 (Chapter 4).
